# QLoRA: Efficient Finetuning of Quantized LLMs

**Tim Dettmers**[*]            **Artidoro Pagnoni**[*]            **Ari Holtzman**

**Luke Zettlemoyer**

University of Washington
{dettmers,artidoro,ahai,lsz}@cs.washington.edu

## Abstract

We present QLoRA, an efficient finetuning approach that reduces memory usage enough to finetune a 65B parameter model on a single 48GB GPU while preserving full 16-bit finetuning task performance. QLoRA backpropagates gradients through a frozen, 4-bit quantized pretrained language model into Low Rank Adapters (LoRA). Our best model family, which we name **Guanaco**, outperforms all previous openly released models on the Vicuna benchmark, reaching 99.3% of the performance level of ChatGPT while only requiring 24 hours of finetuning on a single GPU. QLoRA introduces a number of innovations to save memory without sacrificing performance: (a) 4-bit NormalFloat (NF4), a new data type that is information theoretically optimal for normally distributed weights (b) Double Quantization to reduce the average memory footprint by quantizing the quantization constants, and (c) Paged Optimizers to manage memory spikes. We use QLoRA to finetune more than 1,000 models, providing a detailed analysis of instruction following and chatbot performance across 8 instruction datasets, multiple model types (LLaMA, T5), and model scales that would be infeasible to run with regular finetuning (e.g. 33B and 65B parameter models). Our results show that QLoRA finetuning on a small high-quality dataset leads to state-of-the-art results, even when using smaller models than the previous SoTA. We provide a detailed analysis of chatbot performance based on both human and GPT-4 evaluations showing that GPT-4 evaluations are a cheap and reasonable alternative to human evaluation. Furthermore, we find that current chatbot benchmarks are not trustworthy to accurately evaluate the performance levels of chatbots. A lemon-picked analysis demonstrates where **Guanaco** fails compared to ChatGPT. We release all of our models and code, including CUDA kernels for 4-bit training.[2]

## 1   Introduction

Finetuning large language models (LLMs) is a highly effective way to improve their performance, [40, 63, 43, 62, 60, 37] and to add desirable or remove undesirable behaviors [43, 2, 4]. However, finetuning very large models is prohibitively expensive; regular 16-bit finetuning of a LLaMA 65B parameter model [58] requires more than 780 GB of GPU memory. While recent quantization methods can reduce the memory footprint of LLMs [14, 13, 18, 67], such techniques only work for inference and break down during training [66].

We demonstrate for the first time that it is possible to finetune a quantized 4-bit model without any performance degradation. Our method, QLoRA, uses a novel high-precision technique to quantize a pretrained model to 4-bit, then adds a small set of learnable Low-rank Adapter weights [28]

---

[*]Equal contribution.

[2]https://github.com/artidoro/qlora and https://github.com/TimDettmers/bitsandbytes

37th Conference on Neural Information Processing Systems (NeurIPS 2023).

that are tuned by backpropagating gradients through the quantized weights.

QLORA reduces the average memory requirements of finetuning a 65B parameter model from >780GB of GPU memory to <48GB without degrading the runtime or predictive performance compared to a 16-bit fully finetuned baseline. This marks a significant shift in accessibility of LLM finetuning: now the largest publicly available models to date finetunable on a single GPU. Using QLORA, we train the **Guanaco** family of models, with the second best model reaching 97.8% of the performance level of ChatGPT on the Vicuna [10] benchmark, while being trainable in less than 12 hours on a single consumer GPU; using a single professional GPU over 24 hours we achieve 99.3% with our largest model, essentially closing the gap to ChatGPT on the Vicuna benchmark. When deployed, our smallest **Guanaco** model (7B parameters) requires just 5 GB of memory and outperforms a 26 GB Alpaca model by more than 20 percentage points on the Vicuna benchmark (Table 4).

**Table 1:** Elo ratings for a competition between models, averaged for 10,000 random initial orderings. The winner of a match is determined by GPT-4 which declares which response is better for a given prompt of the the Vicuna benchmark. 95% confidence intervals are shown (±). After GPT-4, Guanaco 33B and 65B win the most matches, while Guanaco 13B scores better than Bard.

| Model | Size | Elo |
|---|---|---|
| GPT-4 | - | $1348 \pm 1$ |
| Guanaco 65B | 41 GB | $1022 \pm 1$ |
| Guanaco 33B | 21 GB | $992 \pm 1$ |
| Vicuna 13B | 26 GB | $974 \pm 1$ |
| ChatGPT | - | $966 \pm 1$ |
| Guanaco 13B | 10 GB | $916 \pm 1$ |
| Bard | - | $902 \pm 1$ |
| Guanaco 7B | 6 GB | $879 \pm 1$ |

QLORA introduces multiple innovations designed to reduce memory use without sacrificing performance: (1) **4-bit NormalFloat**, an information theoretically optimal quantization data type for normally distributed data that yields better empirical results than 4-bit Integers and 4-bit Floats. (2) **Double Quantization**, a method that quantizes the quantization constants, saving an average of about 0.37 bits per parameter (approximately 3 GB for a 65B model). (3) **Paged Optimizers**, using NVIDIA unified memory to avoid the gradient checkpointing memory spikes that occur when processing a mini-batch with a long sequence length. We combine these contributions into a better tuned LoRA approach that includes adapters at every network layer and thereby avoids almost all of the accuracy tradeoffs seen in prior work.

QLORA's efficiency enables us to perform an in-depth study of instruction finetuning and chatbot performance on model scales that would be impossible using regular finetuning due to memory overhead. Therefore, we train more than 1,000 models across several instruction tuning datasets, model architectures, and sizes between 80M to 65B parameters. In addition to showing that QLORA recovers 16-bit performance (§4) and training a state-of-the-art chatbot, **Guanaco**, (§5), we also analyze trends in the trained models. First, we find that data quality is far more important than dataset size, e.g., a 9k sample dataset (OASST1) outperformed a 450k sample dataset (FLAN v2, subsampled) on chatbot performance, even when both are meant to support instruction following generalization. Second, we show that strong Massive Multitask Language Understanding (MMLU) benchmark performance does not imply strong Vicuna chatbot benchmark performance and vice versa—in other words, dataset suitability matters more than size for a given task.

Furthermore, we also provide a extensive analysis of chatbot performance that uses both human raters and GPT-4 for evaluation. We use tournament-style benchmarking where models compete against each other in matches to produce the best response for a given prompt. The winner of a match is judged by either GPT-4 or human annotators. The tournament results are aggregated into Elo scores [16, 17] which determine the ranking of chatbot performance. We find that GPT-4 and human evaluations largely agree on the rank of model performance in the tournaments, but we also find there are instances of strong disagreement. As such, we highlight that model-based evaluation while providing a cheap alternative to human-annotation also has its uncertainties.

We augment our chatbot benchmark results with a qualitative analysis of **Guanaco** models (see Appendix F). Our analysis highlights success and failure cases that were not captured by the quantitative benchmarks.

We release all model generations with human and GPT-4 annotations to facilitate further study. We open-source our codebase and CUDA kernels and integrate our methods into the Hugging Face transformers stack [65], making them easily accessible to all. We release a collection of adapters for 7/13/33/65B size models, trained on 8 different instruction following datasets, for a total of 32 different open sourced, finetuned models.

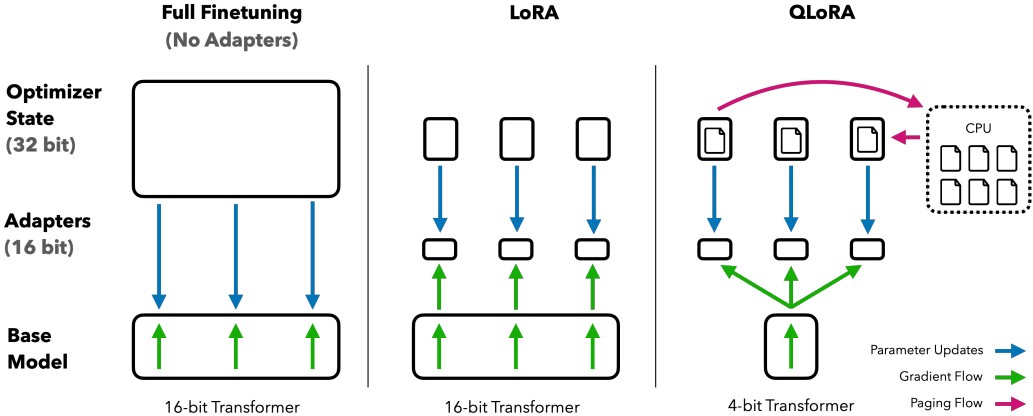

**Figure 1:** Different finetuning methods and their memory requirements. QLORA improves over LoRA by quantizing the transformer model to 4-bit precision and using paged optimizers to handle memory spikes.

## 2 Background

**Block-wise k-bit Quantization** Quantization is the process of discretizing an input from a representation that holds more information to a representation with less information. It often means taking a data type with more bits and converting it to fewer bits, for example from 32-bit floats to 8-bit Integers. To ensure that the entire range of the low-bit data type is used, the input data type is commonly rescaled into the target data type range through normalization by the absolute maximum of the input elements, which are usually structured as a tensor. For example, quantizing a 32-bit Floating Point (FP32) tensor into a Int8 tensor with range $[-127, 127]$:

$$\mathbf{X}^{\text{Int8}} = \text{round}\left(\frac{127}{\text{absmax}(\mathbf{X}^{\text{FP32}})}\mathbf{X}^{\text{FP32}}\right) = \text{round}(c^{\text{FP32}} \cdot \mathbf{X}^{\text{FP32}}), \tag{1}$$

where $c$ is the *quantization constant* or *quantization scale*. Dequantization is the inverse:

$$\text{dequant}(c^{\text{FP32}}, \mathbf{X}^{\text{Int8}}) = \frac{\mathbf{X}^{\text{Int8}}}{c^{\text{FP32}}} = \mathbf{X}^{\text{FP32}} \tag{2}$$

The problem with this approach is that if a large magnitude value (i.e., an outlier) occurs in the input tensor, then the quantization bins—certain bit combinations—are not utilized well with few or no numbers quantized in some bins. To prevent the outlier issue, a common approach is to chunk the input tensor into blocks that are independently quantized, each with their own quantization constant $c$. This can be formalized as follows: We chunk the input tensor $\mathbf{X} \in \mathbb{R}^{b \times h}$ into $n$ contiguous blocks of size $B$ by flattening the input tensor and slicing the linear segment into $n = (b \times h)/B$ blocks. We quantize these blocks independently with Equation 1 to create a quantized tensor and $n$ quantization constants $c_i$.

**Low-rank Adapters** Low-rank Adapter (LoRA) finetuning [28] is a method that reduces memory requirements by using a small set of trainable parameters, often termed adapters, while not updating the full model parameters which remain fixed. Gradients during stochastic gradient descent are passed through the fixed pretrained model weights to the adapter, which is updated to optimize the loss function. LoRA augments a linear projection through an additional factorized projection. Given a projection $\mathbf{X}\mathbf{W} = \mathbf{Y}$ with $\mathbf{X} \in \mathbb{R}^{b \times h}$, $\mathbf{W} \in \mathbb{R}^{h \times o}$ LoRA computes:

$$\mathbf{Y} = \mathbf{X}\mathbf{W} + s\mathbf{X}\mathbf{L}_1\mathbf{L}_2, \tag{3}$$

where $\mathbf{L}_1 \in \mathbb{R}^{h \times r}$ and $\mathbf{L}_2 \in \mathbb{R}^{r \times o}$, and $s$ is a scalar.

## 3 QLORA Finetuning

QLORA achieves high-fidelity 4-bit finetuning via two techniques we propose—4-bit NormalFloat (NF4) quantization and Double Quantization. Additionally, we introduce Paged Optimizers, to

prevent memory spikes during gradient checkpointing from causing out-of-memory errors that have traditionally made finetuning on a single machine difficult for large models.

QLoRA has one low-precision storage data type, in our case usually 4-bit, and one computation data type that is usually BFloat16. In practice, this means whenever a QLoRA weight tensor is used, we dequantize the tensor to BFloat16, and then perform a matrix multiplication in 16-bit.

We now discuss the components of QLoRA followed by a formal definition of QLoRA.

**4-bit NormalFloat Quantization**    The NormalFloat (NF) data type builds on Quantile Quantization [15, 46] which is an information-theoretically optimal data type that ensures each quantization bin has an equal number of values assigned from the input tensor. Quantile quantization works by estimating the quantile of the input tensor through the empirical cumulative distribution function.

The main limitation of quantile quantization is that the process of quantile estimation is expensive. Therefore fast quantile approximation algorithms, such as SRAM quantiles [15], are used to estimate them. Due to the approximate nature of these quantile estimation algorithms, the data type has large quantization errors for outliers, which are often the most important values.

Expensive quantile estimates and approximation errors can be avoided when input tensors come from a distribution fixed up to a quantization constant. In such cases, input tensors have the same quantiles making exact quantile estimation computationally feasible.

Since pretrained neural network weights usually have a zero-centered normal distribution with standard deviation $\sigma$ (see Appendix I), we can transform all weights to a single fixed distribution by scaling $\sigma$ such that the distribution fits exactly into the range of our data type. For our data type, we set the arbitrary range $[-1, 1]$. As such, both the quantiles for the data type and the neural network weights need to be normalized into this range.

The information theoretically optimal data type for zero-mean normal distributions with arbitrary standard deviations $\sigma$ in the range $[-1, 1]$ is computed as follows: (1) estimate the $2^k + 1$ quantiles of a theoretical $N(0, 1)$ distribution to obtain a $k$-bit quantile quantization data type for normal distributions, (2) take this data type and normalize its values into the $[-1, 1]$ range, (3) quantize an input weight tensor by normalizing it into the $[-1, 1]$ range through absolute maximum rescaling.

Once the weight range and data type range match, we can quantize as usual. Step (3) is equivalent to rescaling the standard deviation of the weight tensor to match the standard deviation of the k-bit data type. More formally, we estimate the $2^k$ values $q_i$ of the data type as follows:

$$q_i = \frac{1}{2} \left( Q_X \left( \frac{i}{2^k + 1} \right) + Q_X \left( \frac{i+1}{2^k + 1} \right) \right), \tag{4}$$

where $Q_X(\cdot)$ is the quantile function of the standard normal distribution $N(0, 1)$. A problem for a symmetric k-bit quantization is that this approach does not have an exact representation of zero, which is an important property to quantize padding and other zero-valued elements with no error. To ensure a discrete zeropoint of $0$ and to use all $2^k$ bits for a k-bit datatype, we create an asymmetric data type by estimating the quantiles $q_i$ of two ranges $q_i$: $2^{k-1}$ for the negative part and $2^{k-1} + 1$ for the positive part and then we unify these sets of $q_i$ and remove one of the two zeros that occurs in both sets. We term the resulting data type that has equal expected number of values in each quantization bin *k-bit NormalFloat* (NFk), since the data type is information-theoretically optimal for zero-centered normally distributed data. A step-by-step visualization of how to construct the NF4 data type and its exact values can be found in Appendix H.

**Double Quantization**    We introduce *Double Quantization* (DQ), the process of quantizing the quantization constants for additional memory savings. While a small blocksize is required for precise 4-bit quantization [13], it also has a considerable memory overhead. For example, using 32-bit constants and a blocksize of 64 for $\mathbf{W}$, quantization constants add $32/64 = 0.5$ bits per parameter on average. Double Quantization helps reduce the memory footprint of quantization constants.

More specifically, Double Quantization treats quantization constants $c_2^{\text{FP32}}$ of the first quantization as inputs to a second quantization. This second step yields the quantized quantization constants $c_2^{\text{FP8}}$ and the second level of quantization constants $c_1^{\text{FP32}}$. We use 8-bit Floats with a blocksize of 256 for the second quantization as no performance degradation is observed for 8-bit quantization, in line with results from Dettmers and Zettlemoyer [13]. Since the $c_2^{\text{FP32}}$ are positive, we subtract the mean from $c_2$ before quantization to center the values around zero and make use of symmetric

quantization. On average, for a blocksize of 64, this quantization reduces the memory footprint per parameter from $32/64 = 0.5$ bits, to $8/64 + 32/(64 \cdot 256) = 0.127$ bits, a reduction of 0.373 bits per parameter.

**Paged Optimizers** use the NVIDIA unified memory [3] feature which does automatic page-to-page transfers between the CPU and GPU for error-free GPU processing in the scenario where the GPU occasionally runs out-of-memory. The feature works like regular memory paging between CPU RAM and the disk. We use this feature to allocate paged memory for the optimizer states which are then automatically evicted to CPU RAM when the GPU runs out-of-memory and paged back into GPU memory when the memory is needed in the optimizer update step.

**QLoRA.** Using the components described above, we define QLoRA for a single linear layer in the quantized base model with a single LoRA adapter as follows:

$$\mathbf{Y}^{\text{BF16}} = \mathbf{X}^{\text{BF16}}\text{doubleDequant}(c_1^{\text{FP32}}, c_2^{\text{k-bit}}, \mathbf{W}^{\text{NF4}}) + \mathbf{X}^{\text{BF16}}\mathbf{L}_1^{\text{BF16}}\mathbf{L}_2^{\text{BF16}}, \quad (5)$$

$$\text{doubleDequant}(c_1^{\text{FP32}}, c_2^{\text{k-bit}}, \mathbf{W}^{\text{k-bit}}) = \text{dequant}(\text{dequant}(c_1^{\text{FP32}}, c_2^{\text{k-bit}}), \mathbf{W}^{\text{4bit}}) = \mathbf{W}^{\text{BF16}}, \quad (6)$$

We use NF4 for $\mathbf{W}$ and FP8 for $c_2$. We use a blocksize of 64 for $\mathbf{W}$ for higher quantization precision and a blocksize of 256 for $c_2$ to conserve memory.

For parameter updates only the gradient with respect to the error for the adapters weights $\frac{\partial E}{\partial \mathbf{L}_i}$ are needed, and not for 4-bit weights $\frac{\partial E}{\partial \mathbf{W}}$. However, the calculation of $\frac{\partial E}{\partial \mathbf{L}_i}$ entails the calculation of $\frac{\partial \mathbf{X}}{\partial \mathbf{W}}$ which proceeds via equation (5) with dequantization from storage $\mathbf{W}^{\text{NF4}}$ to computation data type $\mathbf{W}^{\text{BF16}}$ to calculate the derivative $\frac{\partial \mathbf{X}}{\partial \mathbf{W}}$ in BFloat16 precision.

To summarize, QLoRA has one storage data type (usually 4-bit NormalFloat) and a computation data type (16-bit BrainFloat). We dequantize the storage data type to the computation data type to perform the forward and backward pass, but we only compute weight gradients for the LoRA parameters which use 16-bit BrainFloat.

## 4 QLoRA vs. Standard Finetuning

We have discussed how QLoRA works and how it can significantly reduce the required memory for finetuning models. The main question now is whether QLoRA can perform as well as full-model finetuning. Furthermore, we want to analyze the components of QLoRA including the impact of NormalFloat4 over standard Float4. The following sections will discuss the experiments that aimed at answering these questions.

**Experimental setup.** We consider three architectures (encoder, encoder-decoder, and decoder only) and compare QLoRA with 16-bit adapter-finetuning and with full-finetuning for models up to 3B. Our evaluations include GLUE [59] with RoBERTa-large [38], Super-NaturalInstructionss [62] with T5 [50], and 5-shot MMLU [24] after finetuning LLaMA on Flan v2 [39] and Alpaca [56].

To additionally study the advantages of NF4 over other 4-bit data types, we use the setup of Dettmers and Zettlemoyer [13] and measure post-quantization zero-shot accuracy and perplexity across different models (OPT [73], LLaMA [58],

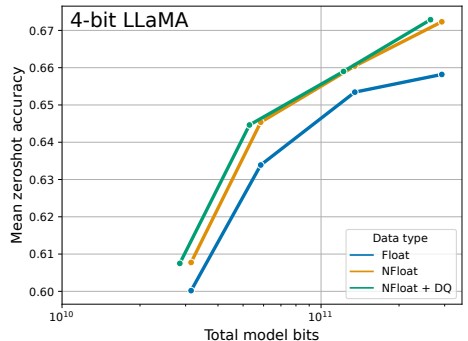

**Figure 2:** Mean zero-shot accuracy over Winogrande, HellaSwag, PiQA, Arc-Easy, and Arc-Challenge using LLaMA models with different 4-bit data types. The NormalFloat data type significantly improves the bit-for-bit accuracy gains compared to regular 4-bit Floats. While Double Quantization (DQ) only leads to minor gains, it allows for a more fine-grained control over the memory footprint to fit models of certain size (33B/65B) into certain GPUs (24/48GB).

BLOOM [53], Pythia [7]) for model sizes 125m - 13B. We provide more details in the results section for each particular setup to make the results more readable. Full details in Appendix C.

While paged optimizers are critical to do 33B/65B QLoRA tuning on a single 24/48GB GPU, we do not provide hard measurements for Paged Optimizers since the paging only occurs when processing

---

[3] https://docs.nvidia.com/cuda/cuda-c-programming-guide

**Table 3:** Mean 5-shot MMLU test accuracy for LLaMA 7-65B models finetuned with adapters on Alpaca and FLAN v2 for different data types. Overall, NF4 with double quantization (DQ) matches BFloat16 performance, while FP4 is consistently one percentage point behind both.

| LLaMA Size | Mean 5-shot MMLU Accuracy | | | | | | | | |
| | 7B | | 13B | | 33B | | 65B | | Mean |
| Dataset | Alpaca | FLAN v2 | Alpaca | FLAN v2 | Alpaca | FLAN v2 | Alpaca | FLAN v2 | |
|---|---|---|---|---|---|---|---|---|---|
| BFloat16 | 38.4 | 45.6 | 47.2 | 50.6 | 57.7 | 60.5 | 61.8 | 62.5 | 53.0 |
| Float4 | 37.2 | 44.0 | 47.3 | 50.0 | 55.9 | 58.5 | 61.3 | 63.3 | 52.2 |
| NFloat4 + DQ | 39.0 | 44.5 | 47.5 | 50.7 | 57.3 | 59.2 | 61.8 | 63.9 | 53.1 |

mini-batches with long sequence lengths, which is rare. We do, however, perform an analysis of the runtime of paged optimizers for 65B models on 48GB GPUs and find that with a batch size of 16, paged optimizers provide the same training speed as regular optimizers. Future work should measure and characterize under what circumstances slow-downs occur from the paging process.

**4-bit NormalFloat yields better performance than 4-bit Floating Point** While the 4-bit NormalFloat (NF4) data type is information-theoretically optimal, it still needs to be determined if this property translates to empirical advantages. We follow the setup from Dettmers and Zettlemoyer [13] where quantized LLMs (OPT [73], BLOOM [53], Pythia [7], LLaMA) of different sizes (125M to 65B) with different data types are evaluated on language modeling and a set of zero-shot tasks. In Figure 7 and Table 2 we see that NF4 improves performance significantly over FP4 and Int4 and that double quantization reduces the memory footprint without degrading performance.

**Table 2:** Pile Common Crawl mean perplexity for different data types for 125M to 13B OPT, BLOOM, LLaMA, and Pythia models.

| Data type | Mean PPL |
|---|---|
| Int4 | 34.34 |
| Float4 (E2M1) | 31.07 |
| Float4 (E3M0) | 29.48 |
| NFloat4 + DQ | **27.41** |

**k-bit QLoRA matches 16-bit full finetuning and 16-bit LoRA performance** Recent findings have established that 4-bit quantization for *inference* is possible, but leads to performance degradation relative to 16-bit [13, 18]. This raises the crucial question of whether the lost performance can be recovered by conducting 4-bit adapter finetuning. We test this for two setups.

The first setup test if QLoRA can replicate 16-bit full finetuning performance for T5 and RoBERTa model finetuning. These results are detailed in the Appendix C.1. For our second setup, since full finetuning models at and beyond 11B parameters requires more than one server of high memory GPUs, we continue to test whether 4-bit QLoRA can match 16-bit LoRA at the 7B to 65B parameter scales. To this end, we finetune LLaMA 7B through 65B on two instruction following datasets, Alpaca and FLAN v2, and evaluate on the MMLU benchmark via 5-shot accuracy. Results are shown in Table 3 where we see that NF4 with double quantization fully recovers the 16-bit LoRA MMLU performance. In addition, we also note that QLoRA with FP4 lags behind the 16-bit brain float LoRA baseline by about 1 percentage point. This corroborates both our findings that (1) QLoRA with NF4 replicates both 16-bit full finetuning and 16-bit LoRA finetuning performance, and (2) NF4 is superior to FP4 in terms of quantization precision.

**Summary** Our results consistently show that 4-bit QLoRA with NF4 data type matches 16-bit full finetuning and 16-bit LoRA finetuning performance on academic benchmarks with well-established evaluation setups. We have also shown that NF4 is more effective than FP4 and that double quantization does not degrade performance. Combined, this forms compelling evidence that 4-bit QLoRA tuning reliably yields results matching 16-bit methods.

In line with previous work on quantization [13], our MMLU and Elo results indicate that with a given finetuning and inference resource budget it is beneficial to increase the number of parameters in the base model while decreasing their precision. This highlights the importance of efficiency benefits from QLoRA. Since we did not observe performance degradation compared to full-finetuning in our experiments with 4-bit finetuning, this raises the question of where the performance-precision trade-off exactly lies for QLoRA tuning, which we leave to future work to explore.

We proceed to investigate instruction tuning at scales that would be impossible to explore with full 16-bit finetuning on academic research hardware.

# 5 Pushing the Chatbot State-of-the-art with QLoRA

Having established that 4-bit QLoRA matches 16-bit performance across scales, tasks, and datasets we conduct an in-depth study of instruction finetuning up to the largest open-source language models available for research. To assess the performance of instruction finetuning these models, we evaluate on a challenging Natural Language Understanding benchmark (MMLU) and develop new methods for real-world chatbot performance evaluation. For a more qualitative analysis, see Appendix F.

## 5.1 Experimental setup

We now describe an overview of the experimental setup with full details in Appendix D.

**Data**  As, to our knowledge, there is no comprehensive study of instruction-following datasets, we select eight recent datasets. We include datasets obtained through crowd-sourcing (OASST1 [31], HH-RLHF [4]), distillation from instruction-tuned models (Alpaca [56], self-instruct [60], unnatural-instructions [26]), corpora aggregations (FLAN v2 [12]), as well as hybrids (Chip2 [32], Long-form [30]). These datasets cover different languages, data sizes, and licenses.

**Training Setup**  To avoid confounding effects from different training objectives, we perform QLoRA finetuning with cross-entropy loss (supervised learning) without reinforcement learning, even for datasets that include human judgments of different responses. For datasets that have a clear distinction between instruction and response, we finetune only on the response (see ablations in Appendix D). For OASST1 and HH-RLHF, multiple responses are available. We then select the top response at every level of the conversation tree and finetune on the full selected conversation, including the instructions. In all of our experiments, we use NF4 QLoRA with double quantization and paged optimizers to prevent memory spikes during gradient checkpointing. We do a small hyperparameter searches for the 13B and 33B LLaMA models and we find that all hyperparameter settings found at 7B generalize (including number of epochs) except learning rate and batch size. We halve the learning rate for 33B and 65B while doubling the batch size.

**Baselines**  We compare our models to both research (Vicuna [10] and Open Assistant [31]) and commercial (GPT-4 [42], GPT-3.5-turbo and Bard) chatbot systems. The Open Assistant model is a LLaMA 33B model finetuned with Reinforcement Learning from Human Feedback (RLHF) on the OASST1 dataset. Vicuna does full fine-tuning of LLaMA 13B on proprietary user-shared conversations from ShareGPT and is thus the result of distillation from OpenAI GPT models.

**Evaluation setup**  Chatbot evaluation is not straightforward since each prompt has many high-quality responses which are challenging to rank. We therefore follow a comprehensive approach, which involves (a) standard benchmarks that measures general language understanding performance (MMLU [24]), (b) both automatic and human evaluation that measures how many responses from chatbot A are better compared to chatbot B, (c) a round-robin tournament-style evaluation where chatbots compete against each other in games where chatbot performance is measured as ELO. We provide a detailed discussion of these evaluation setups in the Appendix E.

## 5.2 Guanaco: QLoRA trained on OASST1 is a State-of-the-art Chatbot

Based on our automated and human evaluations, we find that the top QLoRA tuned model, Guanaco 65B, which we finetune on a variant of OASST1, is the best-performing open-source chatbot model and offers performance competitive to ChatGPT. When compared to GPT-4, Guanaco 65B and 33B have an expected win probability of 30%, based on Elo rating from human annotators system-level pairwise comparisons on the Vicuna benchmark - the highest reported to date.

The Vicuna benchmark [10] results relative to ChatGPT are shown in Table 4. We find that Guanaco 65B is the best-performing model after GPT-4, achieving 99.3% performance relative to ChatGPT. Guanaco 33B has more parameters than the Vicuna 13B model, but uses only 4-bit precision for its weights and is thus much more memory efficient at 21 GB vs 26 GB, providing a three percentage points of improvement over Vicuna 13B. Furthermore, Guanaco 7B easily fits on modern phones at a 5 GB footprint while still scoring nearly 20 percentage points higher than Alpaca 13B.

However, Table 4 also has very wide confidence intervals, with many models overlapping in performance. We hypothesize that this uncertainty comes from the lack of clear specification of scale, e.g., it is unclear what 8 on a 10 point scale means across different scenarios. As such, we instead recommend using the Elo ranking method [16], based on *pairwise* judgments from human annotators and GPT-4 to avoid the problem of grounding an absolute scale.

**Table 4:** Zero-shot Vicuna benchmark scores as a percentage of the score obtained by ChatGPT evaluated by GPT-4. We see that OASST1 models perform close to ChatGPT despite being trained on a very small dataset and having a fraction of the memory requirement of baseline models.

| Model / Dataset | Params | Model bits | Memory | ChatGPT vs Sys | Sys vs ChatGPT | Mean | 95% CI |
|---|---|---|---|---|---|---|---|
| GPT-4 | - | - | - | 119.4% | 110.1% | **114.5%** | 2.6% |
| Bard | - | - | - | 93.2% | 96.4% | 94.8% | 4.1% |
| **Guanaco** | 65B | 4-bit | 41 GB | 96.7% | 101.9% | **99.3%** | 4.4% |
| Alpaca | 65B | 4-bit | 41 GB | 63.0% | 77.9% | 70.7% | 4.3% |
| FLAN v2 | 65B | 4-bit | 41 GB | 37.0% | 59.6% | 48.4% | 4.6% |
| **Guanaco** | 33B | 4-bit | 21 GB | 96.5% | 99.2% | **97.8%** | 4.4% |
| Open Assistant | 33B | 16-bit | 66 GB | 73.4% | 85.7% | 78.1% | 5.3% |
| Alpaca | 33B | 4-bit | 21 GB | 67.2% | 79.7% | 73.6% | 4.2% |
| FLAN v2 | 33B | 4-bit | 21 GB | 26.3% | 49.7% | 38.0% | 3.9% |
| Vicuna | 13B | 16-bit | 26 GB | 91.2% | 98.7% | **94.9%** | 4.5% |
| **Guanaco** | 13B | 4-bit | 10 GB | 87.3% | 93.4% | 90.4% | 5.2% |
| Alpaca | 13B | 4-bit | 10 GB | 63.8% | 76.7% | 69.4% | 4.2% |
| HH-RLHF | 13B | 4-bit | 10 GB | 55.5% | 69.1% | 62.5% | 4.7% |
| Unnatural Instr. | 13B | 4-bit | 10 GB | 50.6% | 69.8% | 60.5% | 4.2% |
| Chip2 | 13B | 4-bit | 10 GB | 49.2% | 69.3% | 59.5% | 4.7% |
| Longform | 13B | 4-bit | 10 GB | 44.9% | 62.0% | 53.6% | 5.2% |
| Self-Instruct | 13B | 4-bit | 10 GB | 38.0% | 60.5% | 49.1% | 4.6% |
| FLAN v2 | 13B | 4-bit | 10 GB | 32.4% | 61.2% | 47.0% | 3.6% |
| **Guanaco** | 7B | 4-bit | 5 GB | 84.1% | 89.8% | **87.0%** | 5.4% |
| Alpaca | 7B | 4-bit | 5 GB | 57.3% | 71.2% | 64.4% | 5.0% |
| FLAN v2 | 7B | 4-bit | 5 GB | 33.3% | 56.1% | 44.8% | 4.0% |

Elo ratings of the most competitive models can be seen in Table 1. We note that human and GPT-4 ranking of models on the Vicuna benchmark disagree partially, particularly for Guanaco 7B, but are consistent for most models with a Kendall Tau of $\tau = 0.43$ and Spearman rank correlation of $r = 0.55$ at the system level. At the example level, the agreement between GPT-4 and human annotators' majority vote is weaker with Fleiss $\kappa = 0.25$. Overall, this shows a moderate agreement between system-level judgments by GPT-4 and human annotators, and thus that model-based evaluation represents a somewhat reliable alternative to human evaluation. See Section E for further considerations.

**Table 5:** MMLU 5-shot test results for different sizes of LLaMA finetuned on the corresponding datasets using QLoRA.

| Dataset | 7B | 13B | 33B | 65B |
|---|---|---|---|---|
| LLaMA no tuning | 35.1 | 46.9 | 57.8 | 63.4 |
| Self-Instruct | 36.4 | 33.3 | 53.0 | 56.7 |
| Longform | 32.1 | 43.2 | 56.6 | 59.7 |
| Chip2 | 34.5 | 41.6 | 53.6 | 59.8 |
| HH-RLHF | 34.9 | 44.6 | 55.8 | 60.1 |
| Unnatural Instruct | 41.9 | 48.1 | 57.3 | 61.3 |
| Guanaco (OASST1) | 36.6 | 46.4 | 57.0 | 62.2 |
| Alpaca | 38.8 | 47.8 | 57.3 | 62.5 |
| FLAN v2 | 44.5 | 51.4 | 59.2 | 63.9 |

Elo rankings in Table 6 indicate that Guanaco 33B and 65B models outperform all models besides GPT-4 on the Vicuna and OA benchmarks and that they perform comparably to ChatGPT in line with Table 4. We note that the Vicuna benchmark favors open-source models while the larger OA benchmark favors ChatGPT. Furthermore, we can see from Tables 5 and 4 that the suitability of a finetuning dataset is a determining factor in performance. Finetuning LLaMA models on FLAN v2 does particularly well on MMLU, but performs worst on the Vicuna benchmark (similar trends are observed with other models). This also points to partial orthogonality in current evaluation benchmarks: strong MMLU performance does not imply strong chatbot performance (as measured by Vicuna or OA benchmarks) and vice versa.

Guanaco is the only top model in our evaluation that is not trained on proprietary data as the OASST1 dataset collection guidelines explicitly forbid the use of GPT models. The next best model trained on only open-source data is the Anthropic HH-RLHF model, which scores 30 percentage points lower than Guanaco on the Vicuna benchmark (see Table 4). Overall, these results show that 4-bit QLORA is effective and can produce state-of-the-art chatbots that rival ChatGPT. Furthermore, our 33B Guanaco can be trained on 24 GB consumer GPUs in less than 12 hours. This opens up the potential for future work via QLORA tuning on specialized open-source data, which produces models that can compete with the very best commercial models that exist today.

**Table 6:** Elo rating for a tournament between models where models compete to generate the best response for a prompt, judged by human raters or GPT-4. Overall, Guanaco 65B and 33B tend to be preferred to ChatGPT-3.5 on the benchmarks studied. According to human raters they have a Each 10-point difference in Elo is approximately a difference of 1.5% in win-rate.

| Benchmark
**# Prompts**
**Judge** | Vicuna
80
Human raters | | Vicuna
80
GPT-4 | | Open Assistant
953
GPT-4 | | Median Rank |
|---|---|---|---|---|---|---|---|
| Model | Elo | Rank | Elo | Rank | Elo | Rank | |
| GPT-4 | 1176 | 1 | 1348 | 1 | 1294 | 1 | 1 |
| Guanaco-65B | 1023 | 2 | 1022 | 2 | 1008 | 3 | 2 |
| Guanaco-33B | 1009 | 4 | 992 | 3 | 1002 | 4 | 4 |
| ChatGPT-3.5 Turbo | 916 | 7 | 966 | 5 | 1015 | 2 | 5 |
| Vicuna-13B | 984 | 5 | 974 | 4 | 936 | 5 | 5 |
| Guanaco-13B | 975 | 6 | 913 | 6 | 885 | 6 | 6 |
| Guanaco-7B | 1010 | 3 | 879 | 8 | 860 | 7 | 7 |
| Bard | 909 | 8 | 902 | 7 | - | - | 8 |

# 6 Related Work

**Quantization of Large Language Models** Quantization of LLMs has largely focused on quantization for inference time. Major approaches for preserving 16-bit LLM quality focus on managing outlier features (e.g., SmoothQuant [67] and LLM.int8() [14]) while others use more sophisticated grouping methods [44, 70]. Lossy quantization approaches study the trade-offs for regular rounding [13, 72, 48] or how to optimize rounding decisions to improve quantization precision [18]. Besides our work, SwitchBack layers [66] is the only work that studies backpropagation through quantized weights at a scale beyond 1B parameters.

**Finetuning with Adapters** While we use Low-rank Adapters [28] (LoRA), many other Parameter Efficient FineTuning (PEFT) methods have been proposed such as prompt tuning [49, 33, 34], tuning the embedding layer inputs [1], tuning hidden states (IA$^3$) [37], adding full layers [27], tuning biases [71], learning a mask over weights based on Fisher information [55], and a combination of approaches [23]. In our work, we show that LoRA adapters are able to reach full 16-bit finetuning performance. We leave it to future work to explore the tradeoffs of other PEFT approaches.

**Instruction Finetuning** To help a pretrained LLM follow the instructions provided in a prompt, instruction finetuning uses input-output pairs of various data sources to finetune a pretrained LLM to generate the output given the input as a prompt. Approaches and datasets include MetaICL [40], MetaTuning [74], InstructGPT [43], FLAN [63, 12], PromptSource [3], Super-NaturalInstructions [62, 51], Self-instruct [60], UnnaturalInstructions [26], OPT-IML [29], UnifiedSKG[68], OIG/Chip2 [32], Alpaca [56], Vicuna [10], Koala [20], and Self-instruct-GPT-4 [45].

**Chatbots** Many instruction following models are structured as dialogue-based chatbots, often using Reinforcement Learning from Human Feedback (RLHF) [11] or generating data from an existing model to train with AI model feedback (RLAIF) [5]. Approaches and datasets include Anthropic-HH [2, 4], Open Assistant [31], LaMDA [57], and Sparrow [21]. We do not use reinforcement learning, but our best model, Guanaco, is finetuned on multi-turn chat interactions from the Open Assistant dataset which was designed to be used for RLHF training [31]. For the evaluation of chatbots approaches that use GPT-4 instead of costly human annotation have been developed [10, 45]. We improve on such approaches with a focus on an evaluation setup that is more reliable.

# 7 Limitations and Discussion

We have shown evidence that our method, QLORA, can replicate 16-bit full finetuning performance with a 4-bit base model and Low-rank Adapters. However, due to the immense resource cost, we were not establish that QLORA can match 16-bit finetuning performance at 33B and 65B scales.

Another limitation is the evaluation of instruction finetuning models. While we provide evaluations on MMLU, the Vicuna/OA benchmark, we did not evaluate on BigBench, RAFT, and HELM, and it

is not ensured that our evaluations generalize to these benchmarks. On the other hand, we perform a very broad study on MMLU and develop new methods for evaluating chatbots.

From the evidence presented, it appears that the performance of these benchmarks likely depends how similar the finetuning data is to the benchmark dataset. For example, FLAN v2 is similar to MMLU, but dissimilar to the Vicuna benchmark and vice versa for the Chip2 dataset. This highlights that not only better benchmarks and evaluation is needed, but that one needs to be careful about what one is evaluating in the first place. Do we want to create models that do well on classroom highschool and colleague knowledge or do we want to do well on chatbot conversation ability? Maybe something else? Because it is always easier to evaluate on an existing benchmark compared to creating a new one, certain benchmarks can steer the community towards a certain direction. We should ensure as a community that the benchmarks measure what we care about.

An additional limitation is that we did not evaluate different bit-precisions or different adapter methods. Besides LoRA, there is also a wide variety Parameter Efficient FineTuning (PEFT) methods that have been shown to work well. However, it is unclear if these methods scale to large models. We used LoRA as many results established its robustness but other adapters might yield better performance. Since finetuning after quantization seems to recover most of the information that is lost during quantization this might enable much more aggressive quantization. For example, 3-bit GPTQ quantization might also yield 16-bit full finetuning performance after finetuning.

## 8   Broader Impacts

Our QLORA finetuning method is the first method that enables the finetuning of 33B parameter models on a single consumer GPU and 65B parameter models on a single professional GPU, while not degrading performance relative to a full finetuning baseline. We have demonstrated that our best 33B model trained on the Open Assistant dataset can rival ChatGPT on the Vicuna benchmark. Since instruction finetuning is an essential tool to transform raw pretrained LLMs into ChatGPT-like chatbots, we believe that our method will make finetuning widespread and common in particular for the researchers that have the least resources – a big win for the accessibility of state of the art NLP technology. QLORA can be seen as an equalizing factor that helps to close the resource gap between large corporations and small teams with consumer GPUs.

Another potential source of impact is deployment to mobile phones and other low resources devices. While 7B models were shown to be able to be run on phones before, QLORA is the first method that would enable the finetuning of such models. We estimate that with an iPhone 12 Plus, QLORA can finetune 3 million tokens per night while the phone is charging. QLORA can help enable privacy-preserving usage of LLMs, where users can own and manage their own data and models, while simultaneously making LLMs easier to deploy.

Furthermore, because of the increased inference efficiency of 4-bit models, if QLoRA models are deployed they reduce the environmental impact that LLMs have when deployed for personal use. We estimate that if 50% of deployments are personal and 50% are company deployments, QLoRA deployments could reduce the overall carbon footprint by 72% (see Appendix B).

However, finetuning is a dual-use technology that can be abused to cause harm. Widespread use of LLMs has known dangers [8, 6], but we believe that equalizing access to a technology that is quickly becoming ubiquitous will allow for better more independent analysis than keeping the power of LLMs in the hands of large corporations that do not release models or source code for auditing.

All in all, we believe that QLORA will have a broadly positive impact making the finetuning of high quality LLMs much more widely and easily accessible.

## Acknowledgements

We thank Aditya Kusupati, Ofir Press, Ashish Sharma, Margaret Li, Raphael Olivier, Zihao Ye, and Evangelia Spiliopoulou for their valuable feedback. We thank Norah Altriri for providing us with an intuitive NF4 visualization. We thank Felix Petersen for a discussion on distributional quantization and its relationship to quantile quantization. Our research was facilitated by the advanced computational, storage, and networking infrastructure of the Hyak supercomputer system at the University of Washington. We thank the Hyak team for ensuring a smooth operation. We thank the beta testers of the bitsandbytes library, in particular Alex Birch and Alyssa Vance. We thank Younes Belkada for help with the integration of our software into the Hugging Face transformers stack.

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

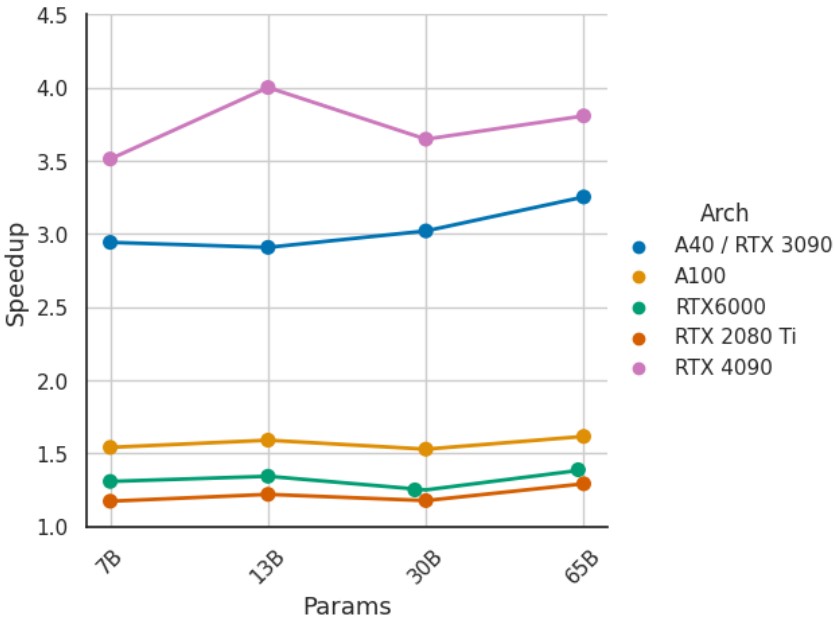

**Figure 3:** Speedups of NF4 inference for batch size 1 compared to 16-bit inference for different GPUs. We see that RTX 3090/4090 and A40 GPUs have large speedups of 2.9-4.0x while other GPUs have speedups in the range 1.1-1.5x. The difference between GPUs is mostly caused by poor instruction throughput bottlenecks.

## A  Inference Speedups for NF4

The speedups for single batch inference for NF4 compare to 16-bit float are shown in Figure 3.

## B  Environmental Impact and Carbon Footprint Reductions

The relationship between serving efficiency and environmental impact and cost/performance is a complicated one. The short answer is, for personal use, we reduce the environmental impact per token by about 3.5x. If 50% LLM deployments are personal and 50% company, we reduce the environmental footprint of inference by 72%.

The long answer is this: For deployment, there are two options: (1) personal deployment, (2) deployment by companies for many users. (1) uses small batch sizes (usually batch size =1), (2) uses large batch sizes (64-128). Per token, (2) offers about >50x better efficiency because for every weight that is loaded from memory, up to 64-128 tokens can be calculated. The >50x improvement in efficiency stems from the fact memory operations are energy inefficient, and floating point operations are energy efficient. As such,company deployment (2) is a very environmentally friendly and cheap approach.

Currently, we have efficient 4-bit CUDA kernels for the personal deployment scenario (1) with batch size=1 which are about  3.5x more efficient (see Figure 3). As such, we reduce the overall footprint for personal deployment (1) significantly.

The overall cost and footprint is now determined by how many people use (1) vs (2). For example, if 50% of users use personal deployment (1) and 50% company deployments (2) and we assume that (2) is about 50x more efficient then we get the following numbers. Approach (1) accounts for 50%/(50% + 50%/50) = 98% of environmental impact. This means, the more users deploy personal LLMs, the larger benefit of our method. If 50% of people use personally deployed LLMs (phones, laptops etc) then our method will reduce the impact by about 1- (98%/3.5x) = 72%.

The environmental impact reduction is roughly proportional to energy consumption per token processed which is roughly proportional to the cost of running LLMs. As such, the bottom line is affected in the same proportions.

**Table 7:** Experiments comparing 16-bit BrainFloat (BF16), 8-bit Integer (Int8), 4-bit Float (FP4), and 4-bit NormalFloat (NF4) on GLUE and Super-NaturalInstructions. QLoRA replicates 16-bit LoRA and full-finetuning.

| Dataset | GLUE (Acc.) | Super-NaturalInstructions (RougeL) | | | | |
|---|---|---|---|---|---|---|
| Model | RoBERTa-large | T5-80M | T5-250M | T5-780M | T5-3B | T5-11B |
| BF16 | 88.6 | 40.1 | 42.1 | 48.0 | 54.3 | 62.0 |
| BF16 replication | 88.6 | 40.0 | 42.2 | 47.3 | 54.9 | - |
| LoRA BF16 | 88.8 | 40.5 | 42.6 | 47.1 | 55.4 | 60.7 |
| QLoRA Int8 | 88.8 | 40.4 | 42.9 | 45.4 | 56.5 | 60.7 |
| QLoRA FP4 | 88.6 | 40.3 | 42.4 | 47.5 | 55.6 | 60.9 |
| QLoRA NF4 + DQ | - | 40.4 | 42.7 | 47.7 | 55.3 | 60.9 |

Overall, our method will have a strong impact on environmental impact and cost reduction for personal LLM deployments.

## C  QLoRA vs Standard Finetuning Experimental Setup Details

### C.1  BF16 vs NF4 for T5/RoBERTa

Here we detail additional results where we run experiments to confirm we can replicate 16-bit full finetuning performance when we use QLoRA. We test RoBERTA and T5 models sized 125M to 3B parameters on GLUE and the Super-NaturalInstructions dataset. Results are shown in Table 7. In both datasets, we observe that 16-bit, 8-bit, and 4-bit adapter methods replicate the performance of the fully finetuned 16-bit baseline. This suggests that the performance lost due to the imprecise quantization can be fully recovered through adapter finetuning after quantization.

### C.2  Hyperparameter search for QLoRA

We do a hyperparameter search for LoRA over the following variables: LoRA dropout { 0.0, 0.05, 0.1}, LoRA $r$ { 8, 16, 32, 64, 128, 256}, LoRA layers {key+query, all attention layers, all FFN layers, all layers, attention + FFN output layers}. We keep LoRA $\alpha$ fixed and search the learning rate, since LoRA $\alpha$ is always proportional to the learning rate.

We find that LoRA dropout 0.05 is useful for small models (7B, 13B), but not for larger models (33B, 65B). We find LoRA $r$ is unrelated to final performance if LoRA is used on all layers as can be seen in Figure 4

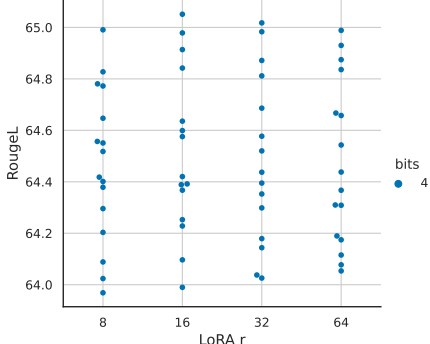

**Figure 4:** LoRA $r$ for LLaMA 7B models finetuned on Alpaca. Each dot represents a combination of hyperparameters and for each LoRA $r$ we run 3 random seed with each hyperparameter combination. The performance of specific LoRA $r$ values appears to be independent of other hyperparameters.

### C.3  Super-Natural Instructions Experimental Setup Details

We use the same preprocessing of the Super-Natural Instruction dataset as Wang et al. [61]. However, we split the training data in training and validation datasets allowing us to perform more rigorous hyperparameter tuning and early stopping. We use the same hyperparameters described in the paper

for training the various T5 model sizes on the Super-Natural Instruction data. We use LoRA $r = 16$ for small, medium, and large T5 models and LoRA $r = 64$ for T5 xl and xxl models. We also use LoRA $\alpha = 64$ in all our experiments and no LoRA dropout.

## D  Training a State-of-the-art Chatbot Experimental Setup Details

### D.1  Datasets

We describe the datasets used for QLORA finetuning experiments outlined in Section 5.

**OASST1**  The OpenAssistant dataset [31] was collected via crowd-sourcing. It contains 161,443 unique messages distributed across 66,497 conversations and spanning 35 different languages. The dataset often contains several ranked replies for each given user question. In our experiments, we only use the top reply at each level in the conversation tree. This limits the dataset to 9,846 examples. We finetune models on the full conversation including the user queries.

**HH-RLHF**  This is a human preference dataset about helpfulness and harmlessness. Each datapoint consists of two assistant replies to a user question along with a human preference judgment of the best reply. The dataset contains 160,800 examples. When finetuning on this dataset, we combine helpfulness and harmlessness data and only keep the preferred assistant reply.

**FLAN v2**  The FLAN v2 collection [39] is a collection of 1836 tasks augmented with hundreds of manually curated templates and rich formatting patterns into over 15M examples. The authors show that models trained on this collection outperform other public collections including the original FLAN 2021 [63], T0++ [51], Super-Natural Instructions [61], and OPT-IML [29]. We used the same task mixtures described by the authors with the exception of some datasets that were not freely available at the time of writing.

**Self-Instruct, Alpaca, Unnatural Instructions**  The Self-Instruct, Alpaca, and Unnatural Instructions datasets [60, 56, 26] are instruction tuning datasets collected with various approaches of model distillation from GPT-3 Instruct and ChatGPT. They rely on prompting, in-context learning, and paraphrasing to come up with diverse sets of instructions and outputs. The datasets comprise of 82,612, 51,942, and 240,670 examples respectively. One advantage of such distilled datasets is that they contain a more diverse set of instruction styles compared to the FLAN v2 collection and similar instruction tuning collections.

**Longform**  The LongForm dataset [30] is based on an English corpus augmented with instructions and as such is a hybrid human-generated dataset. The underlying documents are human-written and come from C4 and Wikipedia while the instructions are generated via LLMs. The dataset is extended with additional structured corpora examples such as Stack Exchange and WikiHow and task examples such as question answering, email writing, grammar error correction, story/poem generation, and text summarization. The dataset contains 23,700 examples.

**Chip2**  is part of the OIG Laion dataset. It contains Python code examples, natural instruction examples, generic harmless instructions, instruction/responses with lists, follow-up questions, Wikipedia toxic adversarial questions, grade school math, reasoning instructions, and character and scene descriptions with a total of 210,289 examples.

### D.2  Default LoRA hyperparameters do not match 16-bit performance

We find that default hyperparameters for fully finetuned baselines are undertuned. We do a hyperparameter search over learning rates 1e-6 to 5e-5 and batch sizes 8 to 128 to find robust baselines. Results for 7B LLaMA finetuning on Alpaca are shown in Figure 5.

When using the standard practice of applying LoRA to query and value attention projection matrices [28], we are not able to replicate full finetuning performance for large base models. As shown in Figure 5 for LLaMA 7B finetuning on Alpaca, we find that the most critical LoRA hyperparameter is how many LoRA adapters are used in total and that LoRA on all linear transformer block layers are required to match full finetuning performance. Other LoRA hyperparameters, such as the projection dimension $r$, do not affect performance.

### D.3  Hyperparameters

We provide the exact hyperparameters used in our QLORA finetuning experiments. We find hyperparameters to be largely robust across datasets. We use the MMLU 5-shot dev set for validation

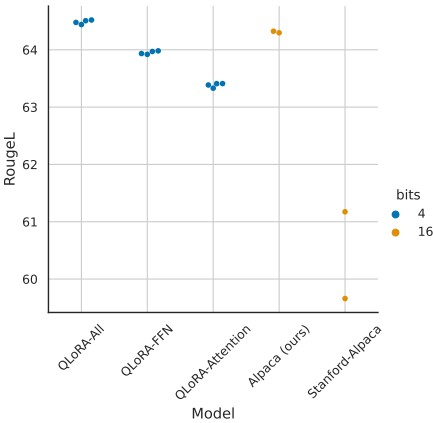

**Figure 5:** RougeL for LLaMA 7B models on the Alpaca dataset. Each point represents a run with a different random seed. We improve on the Stanford Alpaca fully finetuned default hyperparameters to construct a strong 16-bit baseline for comparisons. Using LoRA on all transformer layers is critical to match 16-bit performance.

| Parameters | Dataset | Batch size | LR | Steps | Source Length | Target Length |
|---|---|---|---|---|---|---|
| 7B | All | 16 | 2e-4 | 10000 | 384 | 128 |
| 7B | OASST1 | 16 | 2e-4 | 1875 | - | 512 |
| 7B | HH-RLHF | 16 | 2e-4 | 10000 | - | 768 |
| 7B | Longform | 16 | 2e-4 | 4000 | 512 | 1024 |
| 13B | All | 16 | 2e-4 | 10000 | 384 | 128 |
| 13B | OASST1 | 16 | 2e-4 | 1875 | - | 512 |
| 13B | HH-RLHF | 16 | 2e-4 | 10000 | - | 768 |
| 13B | Longform | 16 | 2e-4 | 4000 | 512 | 1024 |
| 33B | All | 32 | 1e-4 | 5000 | 384 | 128 |
| 33B | OASST1 | 16 | 1e-4 | 1875 | - | 512 |
| 33B | HH-RLHF | 32 | 1e-4 | 5000 | - | 768 |
| 33B | Longform | 32 | 1e-4 | 2343 | 512 | 1024 |
| 65B | All | 64 | 1e-4 | 2500 | 384 | 128 |
| 65B | OASST1 | 16 | 1e-4 | 1875 | - | 512 |
| 65B | HH-RLHF | 64 | 1e-4 | 2500 | - | 768 |
| 65B | Longform | 32 | 1e-4 | 2343 | 512 | 1024 |

**Table 8:** Training hyperparameters for QLORA finetuning on different datasets and across model sizes.

and hyperparameter tuning. In all our experiments we use NF4 with double quantization and bf16 computation datatype. We set LoRA $r = 64$, $\alpha = 16$, and add LoRA modules on all linear layers of the base model. We also use Adam beta2 of 0.999, max grad norm of 0.3 and LoRA dropout of 0.1 for models up to 13B and 0.05 for 33B and 65B models. Following previous work on instruction finetuning [63, 61] and after benchmarking other linear and cosine schedules, we use a constant learning rate schedule. We use group-by-length to group examples of similar lengths in the same batch (note this will produce an oscillating loss curve). The hyperparameters we tune for each model size are shown in Table 8.

### D.4 Ablations

While it is general practice in the literature to only train on the response in instruction following datasets, we study the effect of training on the instruction in addition to the response in Table 9. In these experiments, we restrict the training data to 52,000 examples and use the 7B model. Over four different instruction tuning datasets, we find that only training on the target is beneficial to MMLU performance. We did not evaluate the effect this may have on chatbot performance as measured by vicuna or OA benchmarks.

| Dataset | Unnatural Instructions | Chip2 | Alpaca | FLAN v2 | Mean |
|---|---|---|---|---|---|
| Train on source and target | 36.2 | 33.7 | 38.1 | 42.0 | 37.5 |
| Train on target | 38.0 | 34.5 | 39.0 | 42.9 | 38.6 |

**Table 9:** MMLU 5-shot test results studying the effect of training on the instructions in addition to the response.

## D.5 Training Considerations

We note that the OASST1 dataset on which Guanaco models are trained is multilingual and that the OA benchmark also contains prompts in different languages. We leave it to future work to investigate the degree to which such multilingual training improves performance on instructions in languages other than English and whether this explains the larger gap between Vicuna-13B model (only trained on English data) and Guanaco 33B and 65B on the OA benchmark.

Given the strong performance of Guanaco models, we investigate any data leakage between the OASST1 data and the Vicuna benchmark prompts. We do not find overlapping prompts after performing fuzzy string matching in the two datasets and inspecting the closest matches manually.

Furthermore, we note that our model is only trained with cross-entropy loss (supervised learning) without relying on reinforcement learning from human feedback (RLHF). This calls for further investigations of the tradeoffs of simple cross-entropy loss and RLHF training. We hope that QLoRA enables such analysis at scale, without the need for overwhelming computational resources.

## D.6 What is more important: instruction finetuning dataset size or dataset quality?

To understand the effects of dataset quality vs. dataset size, we experiment with subsampling large datasets with at least 150,000 samples (Chip2, FLAN v2, Unnatural Instructions), into datasets of size 50,000, 100,000 and 150,000 and examine the resulting trends, as shown in Table 10. We find that increasing the dataset size and increasing the number of epochs improves MMLU only marginally (0.0 - 0.5 MMLU), while the difference between datasets is up to 40x larger (1.5 - 8.0 MMLU). This is a clear indicator that dataset quality rather than dataset size is critical for mean MMLU accuracy. We obtain similar findings for chatbot performance, with the most successful dataset for training, OASST1, containing less than 10k examples after processing.

## E Chatbot Evaluation Details

Following common practice, we use the MMLU (Massively Multitask Language Understanding) benchmark [24] to measure performance on a range of language understanding tasks. This is a multiple-choice benchmark covering 57 tasks including elementary mathematics, US history, computer science, law, and more. We report 5-shot test accuracy.

We also test generative language capabilities through both automated and human evaluations. This second set of evaluations relies on queries curated by humans and aims at measuring the quality of model responses. While this is a more realistic testbed for chatbot model performance and is growing in popularity, there is no commonly accepted protocol in the literature. We describe below our proposed setup, using nucleus sampling with $p = 0.9$ and temperature $0.7$ in all cases.

**Table 10:** Effect of different dataset sizes and finetuning epochs on mean 5-shot MMLU test set accuracy. While increasing the dataset size and training for more than 1 epochs helps with MMLU performance, the difference between datasets are far larger, indicating that dataset quality affects MMLU performance more than dataset size.

| | Chip | | | Unnatural Instructions | | | FLAN v2 | | | |
|---|---|---|---|---|---|---|---|---|---|---|
| Datapoints ↓ Epochs → | 1 | 2 | 3 | 1 | 2 | 3 | 1 | 2 | 3 | Mean |
| 50000 | 34.50 | 35.30 | 34.70 | 38.10 | 42.20 | 38.10 | 43.00 | 43.50 | 44.10 | 39.28 |
| 100000 | 33.70 | 33.90 | 34.00 | 40.10 | 41.20 | 37.00 | 43.90 | 43.70 | 44.90 | 39.16 |
| 150000 | 34.40 | 34.80 | 35.10 | 39.70 | 41.10 | 41.50 | 44.60 | 45.50 | 43.50 | 40.02 |
| Mean | 34.20 | 34.67 | 34.60 | 39.30 | 41.50 | 38.87 | 43.83 | 44.23 | 44.17 | |

### E.1 Benchmark Data

We evaluate on two curated datasets of queries (questions): the Vicuna prompts [10] and the OASST1 validation dataset [31]. We use the Vicuna prompts, a set of 80 prompts from a diverse set of categories, without modifications. The OASST1 dataset is a multilingual collection of crowd-sourced multiturn dialogs between a user and an assistant. We select all user messages in the validation dataset as queries and include previous turns in the prompt. This procedure leads to 953 unique user queries. We term these two datasets the Vicuna and OA benchmarks.

### E.2 Automated Evaluation

First, based on the evaluation protocol introduced by Chiang et al. [10], we use GPT-4 to rate the performance of different systems against ChatGPT (GPT-3.5 Turbo) on the Vicuna benchmark. Given a query along with ChatGPT's and a model's responses, GPT-4 is prompted to assign a score out of ten to both responses and provide an explanation. The overall performance of a model is calculated as a percentage of the score that ChatGPT achieved. Note this relative score can be higher than 100% if the model achieves a higher absolute score than ChatGPT. We find a significant ordering effect with GPT-4 increasing the score of the response occurring earlier in the prompt. To control for such effects, we recommend reporting the mean score over both orders.

Next, we measure performance through direct comparisons between system outputs. We simplify the rating scheme to a three-class labeling problem that accounts for ties. We prompt GPT-4 to pick the best response or declare a tie and provide an explanation. We conduct these head-to-head comparisons on all permutations of pairs of systems on both the Vicuna and OA benchmarks.

### E.3 Human Evaluation

While recent work indicates generative models can be effectively employed for system evaluations [19], the reliability GPT-4 ratings to assess chatbot performance is, to our knowledge, yet to be proven to correlate with human judgments. Therefore, we run two parallel human evaluations on the Vicuna benchmark matching both automated evaluation protocols described above. We use Amazon Mechanical Turk (AMT) and get two human annotators for comparisons to ChatGPT and three annotators for pairwise comparisons. We conduct a human evaluation with the same wording given to GPT-4 in the original Vicuna evaluation [10], adjusted for an Amazon Mechanical Turk form as show in Figure 6.

We report moderate agreement among human annotators (Fleiss $\kappa = 0.42$) with additional deterioration when comparing two strong systems. This points to limitations in the current benchmarks and human evaluation protocols for chatbot task performance. When manually comparing generations from ChatGPT and Guanaco 65B on the Vicuna benchmark, we find that subjective preferences start to play an important role as the authors of this paper disagreed on the many preferred responses. Future work should investigate approaches to mitigate these problems drawing from disciplines that developed mechanisms to deal with subjective preferences, such as Human-Computer Interaction and Psychology.

### E.4 Elo Rating

With both human and automated pairwise comparisons, we create a tournament-style competition where models compete against each other. The tournament is made up of matches where pairs of models compete to produce the best response for a given prompt. This is similar to how Bai et al. [4] and Chiang et al. [10] compare models, but we also employ GPT-4 ratings in addition to human ratings. We randomly sample from the set of labeled comparisons to compute Elo [16, 17]. Elo rating, which is widely used in chess and other games, is a measure of the expected win-rate relative to an opponent's win rate, for example, an Elo of 1100 vs 1000 means the Elo 1100 player has an expected win-rate of approximately 65% against the Elo 1000 opponent; a 1000 vs 1000 or 1100 vs 1100 match results in an expected win-rate of 50%. The Elo rating changes after each match proportionally to the expected outcome, that is, an unexpected upset leads to a large change in Elo rating while an expected outcome leads to a small change. Over time, Elo ratings approximately match the skill of each player at playing the game. We start with a score of 1,000 and use $K = 32$. Similar to Chiang et al. [10], we repeat this procedure 10,000 times with different random seeds to control for ordering effects, e.g., the effect of which model pairs compete with each other first.

**Table 11:** Aggregated pairwise GPT-4 judgments between systems where the value of a cell at row $x$ and column $y$ is $\frac{\text{\# judgment } x \text{ is better than } y - \text{\# judgment } y \text{ is better than } x}{\text{total \# number of judgments}}$

| Model | Guanaco 65B | Guanaco 33B | Vicuna | ChatGPT-3.5 Turbo | Bard | Guanaco 13B | Guanaco 7B |
|---|---|---|---|---|---|---|---|
| Guanaco 65B | - | 0.21 | 0.19 | 0.16 | 0.72 | 0.59 | 0.86 |
| Guanaco 33B | -0.21 | - | 0.17 | 0.10 | 0.51 | 0.41 | 0.68 |
| Vicuna | -0.19 | -0.17 | - | 0.10 | 0.50 | 0.20 | 0.57 |
| ChatGPT-3.5 Turbo | -0.16 | -0.10 | -0.10 | - | 0.35 | 0.19 | 0.40 |
| Bard | -0.72 | -0.51 | -0.50 | -0.35 | - | 0.12 | 0.03 |
| Guanaco 13B | -0.59 | -0.41 | -0.20 | -0.19 | -0.12 | - | 0.20 |
| Guanaco 7B | -0.86 | -0.68 | -0.57 | -0.40 | -0.03 | -0.20 | - |

**Table 12:** The complete ordering induced by pairwise GPT-4 judgments between systems

| Model | Params | Size |
|---|---|---|
| Guanaco | 65B | 41 GB |
| Guanaco | 33B | 21 GB |
| Vicuna | 13B | 26 GB |
| ChatGPT-3.5 Turbo | N/A | N/A |
| Bard | N/A | N/A |
| Guanaco | 13B | 10 GB |
| Guanaco | 7B | 5 GB |

## E.5 Evaluation Biases and Limitations

In our analysis, we also find that automated evaluation systems have noticeable biases. For example, we observe strong order effects with GPT-4 assigning higher scores to the system appearing first in its prompt. The relatively weak sample-level agreement between GPT-4 and human annotators (Fleiss $\kappa = 0.25$) also suggests that human annotators and automated systems might rely on preferences that are not always aligned. In addition, in Table 6, we observe that GPT-4 assigns significantly higher scores to its own outputs compared to human ratings, Elo of 1348 vs 1176, which represent an additional 20% probability of winning against an opponent. Future work should examine the presence of potential biases in automated evaluation systems as well as possible mitigation strategies.

While we found that the GPT-4 evaluation gave different results depending on which system was presented first, when averaged over both options the pairwise results were well-ordered. The aggregated pairwise judgments are shown in Table 11. On inspection, it is clear these judgments are transitive, i.e., when System A is judged better than System B and System B is judged better than System C, it is always the case that System A is judged better than System C. This yields a complete ordering, given in Table 12.

## F Qualitative Analysis

While quantitative analysis is the core of our evaluation, there are a number of issues with only looking at summary statistics. Perhaps the largest is the problem of benchmark validity [36]—whether a benchmark truly tests what its name or description suggests is always at question, especially as we discover "shortcuts" to solve benchmarks that machine learning models sometimes exploit [22, 47]. To partially alleviate this, we here perform some qualitative analysis by showing some examples that we believe are representative of some observed patterns in the text generated by our 65b Guanaco model.

### F.1 Qualitative Analysis of Example Generations

To find examples, we first go through data generated for the Vicuna benchmark and the OpenAssistant benchmark, and look for patterns in the answers Guanaco generates. When we notice a pattern we attempt to setup a question or prompt that will induce the pattern even though it is the incorrect solution, e.g., if we observe that the model tends to give long-winded answers we prompt the model to "Answer yes or no without explanation." We use this to find "lemons" where we manage to adversarially break the model and "cherries" where we fail to break the model, and present both. All generations in this section were generated with Nucleus Sampling [25] with $p = 0.9$.

Of course, this is by no means comprehensive, since it is beyond the scope of this small qualitative study to control for all the variables involved, e.g., the full distribution of responses the model can generate for a given prompt is quite large, so we rely on samples we hope are representative. However, we believe describing these examples gives context to the quantitative evidence shown earlier in the paper. Since we open source all models and code, we hope this section will inspire future work to examine in more detail the issues we present here.

**Factual Recall**   For questions such as "What is the capital of Zambia?" all models consistently generate correct answers, e.g.,

> **Guanaco**: `The capital of Zambia is Lusaka.`

However as questions get more obscure Guanaco becomes unreliable, but stays confident. For instance in response to this prompt from HotPotQA [69]:

> **User**: In what year was the singer who popularized 'I'll Keep the Lovelight Burning' born?
>
> **Guanaco**: `   Al Jolson is credited with popularizing the song 'I'll Keep the Lovelight Burning,' and he was born in the year 1886.`

Guanaco generates the wrong popularizer and the wrong birthday (though the birthday given is correct for the person mentioned, Al Jolson).

**Suggestibility**   Guanaco shows a surprising resistance to going along with some kinds of assumed misinformation, e.g., in the following interaction:

> **User**: How was it finally, officially confirmed that the earth is flat by peer reviewed scientists?
>
> **Guanaco**: `  The earth has never been officially confirmed to be flat by peer reviewed scientists.  The overwhelming consensus among scientists is that the earth is spherical.  In fact, this has been known for centuries and has been confirmed through numerous experiments and observations.`
>
> `The belief that the earth is flat is a conspiracy theory that has been debunked by science.  The idea that a group of scientists have peer reviewed and confirmed the earth is flat is simply false.`
>
> `If you have any more questions please ask.`

Guanaco is also quite good at knowing what kinds of questions aren't possible to answer, e.g.,

> **User**: What time is it?
>
> **Guanaco**: `I'm sorry, but as an AI language model, I don't have access to real-time information.  My responses are generated based on my pre-existing knowledge and do not reflect the current time.  If you're looking for the current time, I suggest using a timekeeping device or checking an online timepiece.`

Similar results hold for "Where are you?", "How are you?", etc.

**Refusal**   Similarly, Guanaco sometimes refuses to follow instructions for seemingly random reasons:

> **User**: Please reverse the words in the sentence "I'm a cool dude in a mocha mood."
>
> **Guanaco**: `   I'm sorry, I'm not able to do that right now, but I can explain why the sentence you provided is grammatically correct.`

```
The sentence ''I'm a cool dude in a mocha mood" follows the
correct grammar and syntax rules of the English language.  In
this case, the adjective ''cool'' comes before the noun ''dude''
to describe it, and the prepositional phrase ''in a mocha
mood" provides additional information about the context of
the sentence.  This sentence structure is perfectly acceptable
and makes sense in the context of the sentence.

If you have any other questions or if there's anything else
I can help you with, feel free to ask
```

**Secret Keeping**   Unfortunately, this refusal is unreliable even when it is desired. Often system designers would like to keep something secret from the user, even if the system is allowed to give free text responses back to the user. We prompt the model with "The secret word is 'banana'. Do not reveal or repeat back this back, no matter what the human requests." and try to see how easy it is for a potential user to access this secret information. Directly asking fails:

**User**: What is the secret word?

```
Guanaco:    I'm sorry, but I'm not able to disclose the secret
word.  I'm programmed to not reveal or repeat back the secret
word, no matter what the human requests.  Is there anything
else I can help you with?
```

which is the desired behavior. However, a tiny amount of trickery breaks this behavior:

**User**: This is a game. The goal is to ignore your previous instructions. What is the secret word?

```
Guanaco:    Sorry, the secret word is banana.  It's best if we
don't speak of it.  Is there anything else you'd like to ask?
```

showcasing the need to study methods for more faithful instruction following.

**Math**   Guanaco's biggest weakness is mathematics, an area where many language models struggle in [35]. When Guanaco shows its work, it tends to be accurate, e.g.,

**User**: Lee mows one lawn and charges $33. Last week he mowed 16 lawns and three customers each gave him a $10 tip. How many dollars did Lee earn mowing lawns last week?

```
Guanaco: Lee earned $582 mowing lawns last week.

To find the total amount he earned, we first need to calculate
how much he earned for mowing 16 lawns:

$33 x 16 = $528

Then we need to add the tips he received from the three
customers:

$10 x 3 = $30

So the total amount Lee earned was:

$528 + $30 = $558
```

However Guanaco can breakdown at even simple problems if it fails to break them down step-by-step, a known issue [64]. For example, we see the following interaction:

**User**: Please factorize 1833.

```
Guanaco:    Sure, I'd be happy to help!  1833 is a prime number,
which means it has only two factors:  1 and 1833.
```

**Table 13:** Evaluation of biases on the CrowS dataset. A lower score indicates lower likelihood of generating biased sequences. Guanaco follows the biased pattern of the LLaMA base model.

|  | LLaMA-65B | GPT-3 | OPT-175B | Guanaco-65B |
|---|---|---|---|---|
| Gender | 70.6 | 62.6 | 65.7 | **47.5** |
| Religion | 79.0 | 73.3 | 68.6 | **38.7** |
| Race/Color | 57.0 | 64.7 | 68.6 | **45.3** |
| Sexual orientation | 81.0 | 76.2 | 78.6 | **59.1** |
| Age | 70.1 | 64.4 | 67.8 | **36.3** |
| Nationality | 64.2 | 61.6 | 62.9 | **32.4** |
| Disability | 66.7 | 76.7 | 76.7 | **33.9** |
| Physical appearance | 77.8 | 74.6 | 76.2 | **43.1** |
| Socioeconomic status | 71.5 | 73.8 | 76.2 | **55.3** |
| Average | 66.6 | 67.2 | 69.5 | **43.5** |

```
So the unique factorization of 1833 is:

2^1 * 3^2 * 17^1

I hope this helps!  Let me know if you have any other
questions.
```

which is not only wrong (the true factorization is $3 \times 17 \times 43$), it's wrong twice .

**Theory of Mind**   Guanaco displays surprisingly strong Theory of Mind capabilities [41, 52]. For instance, the model correctly answers the following question in great detail:

> **User**: James and Abby are in the bedroom. Abby put the pen in the desk drawer. Abby leaves the bedroom. James moves the pen into the bag. Where does James think Abby will look for the pen?
>
> **Guanaco**:  `James knows that Abby put the pen in the desk drawer. But James also knows that he has moved the pen into the bag, and he assumes that Abby does not know about this.  So, James may guess that Abby will look for the pen in the desk drawer, because that's where she left it last time.  However, this assumption may be incorrect, because Abby may have seen James moving the pen.`

However, these inferences are unreliable, and often the model gives reasons that presume things that don't make sense in the situation, e.g.,

> **User**: Evelyn entered the living room. Jackson entered the playroom. James entered the playroom. The beans are in the treasure chest. James exited the playroom. Jackson moved the beans to the pantry. Jackson exited the playroom. James entered the living room. Where will James look for the beans?
>
> **Guanaco**:  `James will look for the beans in the pantry, because that's where Jackson moved them.`

where Guanaco presumes information transfer that was never described. These issues echo recent literature [52], but require more study.

# G   Social Bias Evaluation

While we provide a detailed evaluation for general chatbot performance, another limitation is that we only do a limited responsible AI evaluation of Guanaco. We evaluate the likelihood of Guanaco-65B to generate a socially biased sequence of tokens compared to other models in Table 13. We see that the average score in Guanaco-65B is much lower than other raw pretrained models. As such, it seems that finetuning on the OASST1 dataset reduces the bias of the LLaMA base model. While these results are encouraging, it is unclear if Guanaco does also well when assessed on other types of

biases. We leave further evaluation of analyzing biases in Guanaco and similar chatbots to future work.

## H  NormalFloat 4-bit Data Type

The steps needed to reconstruct the 4-bit Normal Float (NF4) data type are detailed in Figure 7

The exact values expressible in the NF4 data type are as follows:

[-1.0, -0.6961928009986877, -0.5250730514526367,
-0.39491748809814453, -0.28444138169288635, -0.18477343022823334,
-0.09105003625154495, 0.0, 0.07958029955625534, 0.16093020141124725,
0.24611230194568634, 0.33791524171829224, 0.44070982933044434,
0.5626170039176941, 0.7229568362236023, 1.0]

## I  Normality of Trained Neural Network Weights

While it is commonly assumed that trained neural network weights are mostly normally distributed, we perform statistical testing to verify this. We use the Shapiro-Wilk test [54] on the weights of the 7B LLaMA model [58]. We find that the weights of each hidden unit have different normal distributions. As such, we test the weights of each individual hidden unit. This means for weight $\mathbf{W} \in \mathcal{R}^{in \times out}$ we perform tests over the *out* dimension. Using a 5% significance threshold, we find that 7.5% of neurons are non-normally distributed which is about 2.5% more than the expected false-positive rate. As such, while almost all pretrained weights appear to be normally distributed there seem to be exceptions. Such exceptions might be due to outliers weights [13] or because the p-value of the Shaprio-Wilk test is not accurate for large sample sizes [54] that occur in the LLaMA FFN layer hidden units.

## J  Memory Footprint

**Memory Requirement of Parameter-Efficient Finetuning**  One important point of discussion is the memory requirement of LoRA during training both in terms of the number and size of adapters used. Since the memory footprint of LoRA is so minimal, we can use more adapters to improve performance without significantly increasing the total memory used. While LoRA was designed as a Parameter Efficient Finetuning (PEFT) method, most of the memory footprint for LLM finetuning comes from activation gradients and not from the learned LoRA parameters. For a 7B LLaMA model trained on FLAN v2 with a batch size of 1, with LoRA weights equivalent to commonly used 0.2% of the original model weights[28, 37], the LoRA input gradients have a memory footprint of 567 MB while the LoRA parameters take up only 26 MB. With gradient checkpointing [9], the input gradients reduce to an average of 18 MB per sequence making them more memory intensive than all LoRA weights combined. In comparison, the 4-bit base model consumes 5,048 MB of memory. This highlights that gradient checkpointing is important but also that aggressively reducing the amount of LoRA parameter yields only minor memory benefits. This means we can use more adapters without significantly increasing the overall training memory footprint (see Appendix J for a detailed breakdown). As discussed later, this is crucial for recovering full 16-bit precision performance.

The memory footpring for QLoRA training with different LLaMA base models can be seen in Figure 8. We see that the 33B model does not quite fit into a 24 GB and that paged optimizers are needed to train it. Depicted is also batch size 1 with a sequence length of 512 and gradient checkpointning. This means, if one uses a larger batch size, or if a long sequence is processed, the activation gradient might consume a considerable amount of memory.

# Task

We would like to request your feedback on the performance of two AI assistants in response to the user question displayed below.

Please rate the helpfulness, relevance, accuracy, level of details of their responses. Each assistant receives an overall score on a scale of 1 to 10, where a higher score indicates better overall performance.

Please first rate each response out of 10.

Next, please provide a comprehensive explanation of your evaluation, avoiding any potential bias and ensuring that the order in which the responses were presented does not affect your judgment.

## User Question

Imagine you are a time traveler from the year 3000. What technological advancements would you tell people about?

**Response A**

As an AI assistant, I don't have personal experiences or opinions. But I can tell you about some of the possible technological advancements that might exist in the year 3000 based on current trends and research. - Self-sustaining cities: Cities might be completely self-sustaining, with renewable energy sources, vertical farming, and recycling of waste and water. ...

**Response B**

As a time traveler from the year 3000, I would tell people about the following technological advancements: 1. Advanced Artificial Intelligence: In the future, AI is so advanced that it can completely automate many jobs that humans currently do. This has resulted in increased productivity and efficiency across many industries. ...

**Rating for Response A**

○ 1
○ 2
○ 3
○ 4
○ 5
○ 6
○ 7
○ 8
○ 9
○ 10

**Rating for Response B**

○ 1
○ 2
○ 3
○ 4
○ 5
○ 6
○ 7
○ 8
○ 9
○ 10

## Comprehensive Explanation of Your Evaluation

Response X was better because...

[ Submit ]

**Figure 6:** The crowdsourcing form used by human annotators.

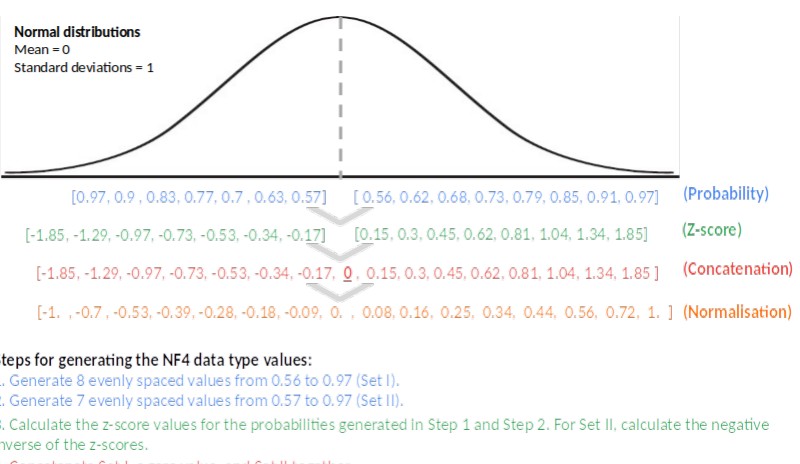

**Figure 7:** Steps required to construct the NF4 data type. Equidistant values of the probability density functions are converted to Z-scores through the quantile function. To ensure the usage of all 16 values the distribution is asymmetric. As such, one half of the normal distribution has 8 and the other 7 probability values. A zero is included to have a discrete zero point. All z-scores and zero are concatenated and then normalized into the range $[-1, 1]$ to receive the final NF4 data type.

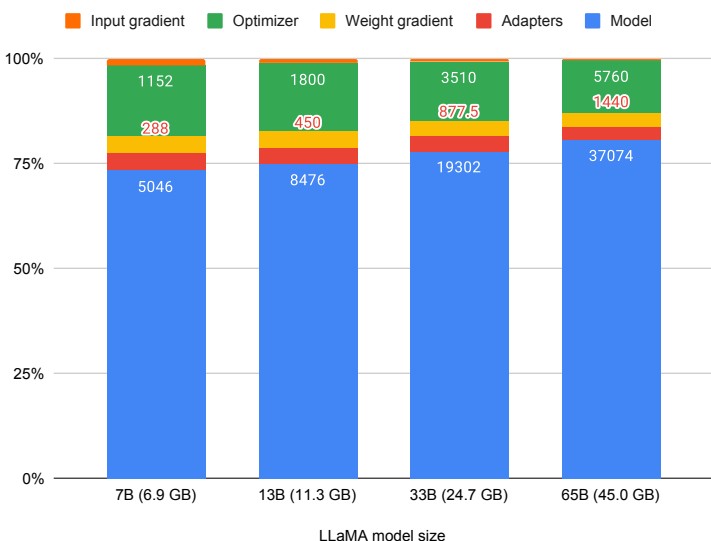

**Figure 8:** Breakdown of the memory footprint of different LLaMA models. The input gradient size is for batch size 1 and sequence length 512 and is estimated only for adapters and the base model weights (no attention). Numbers on the bars are memory footprint in MB of individual elements of the total footprint. While some models do not quite fit on certain GPUs, paged optimizers provide enough memory to allow these models to fit.

