# OpenReview forum: "QLoRA: Efficient Finetuning of Quantized LLMs"
_NeurIPS.cc/2023/Conference — NeurIPS 2023 oral_

### Official Review · Reviewer_enuE · 2023-07-03

**Soundness:** 3 good
**Presentation:** 3 good
**Contribution:** 4 excellent
**Rating:** 8
**Confidence:** 4

**Summary:**

This work pushes the limit of memory-efficient fine-tuning of LLMs by using LoRA over 4-bit quantized LLM. The authors propose NF4, a new data type for normal data. Additionally, efficiency tricks have been proposed, including double quantization, which promotes the quantization of the quantization constant and paged optimizer for long-sequence mini-batches. Combined, QLoRA is able to fine-tune 65B LLMs on a single GPU.

**Strengths:**

1. The fine-tuned models are thoroughly evaluated, and the ablations in the appendix complement the main paper.
2. This work attempts to solve a broader problem in the deep learning community and can do so successfully, unlocking the application of QLoRA on other large-scale deep learning models.
3. The evaluation is thorough with the exact specifications of the compared methods, dataset, hyperparameters and base models.

**Weaknesses:**

1. No serious weakness apart from minor revisions (see questions)

**Questions:**

1. Can the authors re-think the naming convention of double quantization, for example, I had to re-read equations 4 and 5 for better understanding.
2. Can the structure of NF4 data type be directly plugged into 8-bit and higher precision data-types for improved performance, if yes is there any experiment to show the prowess of the NFX data types?

**Limitations:**

Yes

---

> ### Author Rebuttal · Authors · 2023-08-08
>
> **Q1: Can the authors re-think the naming convention of double quantization, for example, I had to re-read equations 4 and 5 for better understanding.**
>
> **A1**: Thank you, we got the same feedback after circulating our draft for review after submission. Since then, this section received a major overhaul and should be much more readable. We mainly focused on an easier readable text and brought the equations closer to double quantization. We made the convention slightly clearer by emphasizing the quantization constants as a target for double quantization.
>
>
> **Q2: Can the structure of NF4 data type be directly plugged into 8-bit and higher precision data-types for improved performance, if yes is there any experiment to show the prowess of the NFX data types?**
>
> **A2**: Yes, NormalFloats can easily be defined for any bit-width. However, with 16-bit inputs and 8-bit weights there is little to no degradation with any data type. Degradation starts to happen at around 6-bit weights and then improved data types show their first effect. When we use NF6, we already see some advantages over FP6. These results improve the fewer bits we have.
> So NF6 advantages over other 6-bit data types are only minor but still significant. These increase with 5 bits, become a sizable advantage at 4-bit (as detailed in the paper), and advantages are very large at 3-bit, but instabilities are a major problem. This means, NF3 has a major advantage over FP3, but for both data types the quantization can fail sporadically leading to random performance. While NF3 certainly is a big improvement over FP3, additional methods are needed to stabilize 3-bit quantizations. We are currently working on this challenging problem.

---

### Official Review · Reviewer_uL2c · 2023-07-06

**Soundness:** 3 good
**Presentation:** 3 good
**Contribution:** 4 excellent
**Rating:** 7
**Confidence:** 5

**Summary:**

The problem addressed in this paper is the high cost associated with fine-tuning large models, such as the standard fp16 LLaMA65B model, which requires over 780G of memory for parameter fine-tuning.

The proposed solution is QLoRA, which consists of three main components: NF4 Quantization (4-bit quantization), Double Quantization, Paged Optimizers.

**Strengths:**

The authors have open-sourced all the training code, which will benefit the community in further researching Large Language Models (LLMs). This paper will have significant implications for the community.

The motivation behind this paper is very clear.

The techniques proposed are innovative.

This paper is well-written.

**Weaknesses:**

The authors are encouraged to discuss the inference efficiency of QLoRA.

**Questions:**

see Weaknesses

**Limitations:**

See Weaknesses

---

> ### Author Rebuttal · Authors · 2023-08-08
>
> **Q1: The authors are encouraged to discuss the inference efficiency of QLoRA.**
>
> **A1**: For large models like LLaMA v1 and v2, matrix multiplication accounts for about 95% of all floating point operations. As such, accelerating 4-bit matrix multiplication accelerates inference speed. Since our submission we implemented highly efficient CUDA inference kernels for 4-bit matrix multiplication (any data type) for batch size one and compared it to 16-bit NVIDIA matrix multiplication performance. These implementations are optimized for RTX 4090 and A40 GPUs. Below the speedups for QLoRA inference. We see strong speedups between 3-4x compared to 16-bit, making inference for QLoRA highly efficient. We will add these data to the appendix of the paper. Please see reviewer hHMr (A1) for further information about inference efficiency.
>
> | GPU      | Storage type | Compute type | Size | Speedup vs 16-bit |
> |----------|--------------|--------------|------|-------------------|
> | RTX 4090 |      NF4     |     BF16     |  7B  |       3.51x       |
> | RTX 4090 |      NF4     |     BF16     |  13B |       4.00x       |
> | RTX 4090 |      NF4     |     BF16     |  30B |       3.65x       |
> | RTX 4090 |      NF4     |     BF16     |  65B |       3.81x       |
> | A40      |      NF4     |     BF16     |  7B  |       2.94x       |
> | A40      |      NF4     |     BF16     |  13B |       2.91x       |
> | A40      |      NF4     |     BF16     |  30B |       3.02x       |
> | A40      |      NF4     |     BF16     |  65B |       3.25x       |

---

### Official Review · Reviewer_pPZQ · 2023-07-07

**Soundness:** 3 good
**Presentation:** 2 fair
**Contribution:** 3 good
**Rating:** 7
**Confidence:** 3

**Summary:**

This paper proposes a method for efficient finetuning large language models. To address the main bottleneck from memory, the authors quantize the model to 4-bit along with standard LoRA adapter. With the proposed method, the authors demonstrate the possibility of finetuning large language model on single GPU with competitive performance.

**Strengths:**

The main strength lies in the system this paper provides. Quantization and adapter fine-tuning are both standard techniques when it comes to efficiency. However, this paper opens up the possibility of finetuning large language model under reasonable cost. Such possibility  is important for the community to study large language model and pursue new research ideas.

**Weaknesses:**

From Table 3, we observe the performance gap increase the model size increase. This could be concerning, especially when the largest model is 11B.  It would be helpful for the community to understand when or to what scale does the proposed method works.

**Questions:**

(1) Quantization and efficient finetuning are generally speaking orthogonal techniques. Is there any specific reason for the choice of LoRA. Can the proposed method be applied on top of other efficient finetuning method?

(2) The authors mention paged optimizer as a major contribution. Can the authors provide more details on the improvement resuling from the paged optimizer?

**Limitations:**

The authors provide discussion on the limitation and potentially negative impact.

---

> ### Author Rebuttal · Authors · 2023-08-07
>
> **Q1: From Table 3, we observe the performance gap increase the model size increase. This could be concerning, especially when the largest model is 11B.[...] helpful [...] to understand when or to what scale does the proposed method works.**
>
> **A1**: We were not able to run a replication for 11B since it consumed too much memory. Furthermore, we find that our replication results are slightly lower, since we use a validation set rather than optimizing the test set as done with the baseline. Thirdly, we find that the 11B model is particularly unstable and does not train easily. This is a common features for T5 models which is the only huggingface model where attention is fully run in FP32 because otherwise its too unstable. All these features made it difficult to obtain the performance of the base model with QLoRA. From other results, we see that QLoRA scales fine to 65B parameters and as such, the deviation from this result stems mostly from the particular instabilities of the T5 model — in particular the 11B T5 model.
>
> **Q2: Quantization and efficient finetuning are generally speaking orthogonal techniques. Is there any specific reason for the choice of LoRA. Can the proposed method be applied on top of other efficient finetuning method?**
>
> **A2:**  In short, we used LoRA due to its robustness, but any other method could be used with our 4-bit quantized base model.
> Our quantization approach is compatibility with any parameter efficient finetuning (PEFT) method. However, we chose LoRA for two reasons: (1) it has been used in many projects with great results, (2) a colleague ran an extensive evaluation of all different PEFT methods and found LoRA was one of two only methods that worked well for finetuning if rigorously evaluated (most PEFT methods are evaluated on GLUE+RoBERTA which is am experiments setup that has many issues). As such, LoRA was an obvious choice for us. While we wanted to also try other methods, we did not have enough computational resources to investigate other methods.
>
> **Q3: The authors mention paged optimizer as a major contribution. Can the authors provide more details on the improvement resuling from the paged optimizer?**
>
> **A3**:
> Paged optimizers prevent memory spikes from mini-batches with long sequence lengths which can cause out of memory errors. While it is possible to restart a model after it crashed and skip the long mini-batch it is usually very difficult to say how much data will be skipped or how often a model will crash. This is so, because before training, it is often very difficult to calculate the exact training memory footprint of a training run without running the model itself.
>
> Paged optimizers solve this issue by lazily preventing memory spikes through offloading to CPU memory temporarily. This eliminates out of memory errors and also makes sure no data is skipped. As such, with paged optimizers, the user knowns even before training that the model will not crash due to out of memory errors and all data will be seen. This enormously simplifies finetuning under constrained memory settings which are common when researchers with few GPUs need to finetune large models for their research. We think this makes paged optimizers a very significant contribution.

---

### Official Review · Reviewer_hHMr · 2023-07-11

**Soundness:** 4 excellent
**Presentation:** 4 excellent
**Contribution:** 4 excellent
**Rating:** 9
**Confidence:** 5

**Summary:**

This article provides a novel way to finetune 4-bit LLMs for efficient finetuning and serving. QLORA, the proposed method, uses a number of innovations to save memory without sacrificing performance. The authors show that QLORA can be used to finetune a 65B parameter model on a single 48GB GPU, and that it performs well in academic and instruction finetune datasets. QLORA is a significant contribution to the field of LLM finetuning and quantization.



**Strengths:**

1. The QLORA paper conducted extensive evaluations on a wide range of datasets, including both academic datasets and the more recent instruction-following datasets.
2. QLORA introduced a novel data type called 4-bit NormalFloat (NF4) for quantization.
3. QLORA can finetune 65B parameter models on a single 48GB GPU, which is significantly more efficient than traditional finetuning methods.
4. QLORA provides good flexibility, as it supports a variety of LLMs, including LLaMA and T5. This suggests that the method can be used to finetune models of different sizes, from small models to large models.
5. QLORA is open sourced with all of the models and code, making it easy for others to use and reproduce the results.




**Weaknesses:**

I wish the authors had discussed the impact of their work on the serving cost of large language models (LLMs). LLM finetuning is relatively inexpensive, and it is good that this work makes it even more affordable. However, the serving cost (in terms of energy consumption and carbon emissions) is much higher than the pretraining cost, especially when there are hundreds of millions of users. This work has the potential to significantly reduce the serving cost of LLMs, which would be a major benefit for both the environment and the bottom line.

Here are some specific points that could be discussed:

a. The authors could estimate the potential savings in energy consumption and carbon emissions that could be achieved by using their method to serve LLMs in 4-bit rather than 8-bit or 16 bit.
b. They could also discuss the potential impact of their work on the economics of LLMs. For example, if the serving cost is significantly reduced, it could make LLMs more affordable for businesses and organizations.
c. Finally, the authors could discuss the challenges that still need to be addressed in order to make their method practical for large-scale deployment.




**Questions:**

1. Can you please replace figure-3 with the table of numbers for Winogrande, HellaSwag, PiQA, Arc-Easy, and Arc-C?

2. Can you please include the ablation studies on the learning-rate / schedule /hyperparameters that you have done in the appendix? Do all the finetuning experiments shared the same recipt?


**Limitations:**

The authors discussed the limitations well in the paper.

---

> ### Author Rebuttal · Authors · 2023-08-07
>
> **Q1: I wish the authors had discussed the impact of their work on the serving cost of large language models (LLMs) [...] This work has the potential to significantly reduce the serving cost of LLMs, which would be a major benefit for both the environment and the bottom line.**
>
> **A1**:
>  The relationship between serving efficiency and environmental impact and cost/performance is a complicated one. The short answer is, for personal use, we reduce the environmental impact per token by about 3.5x. If 50% LLM deployments are personal and 50% company, we reduce the environmental footprint of inference by 72%.
>
> The long answer is this: For deployment, there are two options: (1) personal deployment, (2) deployment by companies for many users. (1) uses small batch sizes (usually batch size =1), (2) uses large batch sizes (64-128). Per token, (2) offers about >50x better efficiency because for every weight that is loaded from memory, up to 64-128 tokens can be calculated. The >50x improvement in efficiency stems from the fact memory operations are energy inefficient, and floating point operations are energy efficient. As such,company deployment (2) is a very environmentally friendly and cheap approach.
>
> Currently, we have efficient 4-bit CUDA kernels for the personal deployment scenario (1) with batch size=1 which are about ~3.5x more efficient (see table below). As such, we reduce the overall footprint for personal deployment (1) significantly.
>
> The overall cost and footprint is now determined by how many people use (1) vs (2). For example, if 50% of users use personal deployment (1) and 50% company deployments (2) and we assume that (2) is about 50x more efficient then we get the following numbers. Approach (1) accounts for 50%/(50% + 50%/50) = 98% of environmental impact.
> This means, the more users deploy personal LLMs, the larger benefit of our method. If 50% of people use personally deployed LLMs (phones, laptops etc) then our method will reduce the impact by about 1- (98%/3.5x) =  72%.
>
> The environmental impact reduction is roughly proportional to energy consumption per token processed which is roughly proportional to the cost of running LLMs. As such, the bottom line is affected in the same proportions.
>
> Overall, our method will have a strong impact on environmental impact and cost reduction for personal LLM deployments.
>
> Table showing increased speedups / efficiency gains of our approach compared to 16-bit inference:
>  GPU      | Storage type | Compute type | Size | Speedup vs 16-bit |
> |----------|--------------|--------------|------|-------------------|
> | RTX 4090 |      NF4     |     BF16     |  7B  |       3.51x       |
> | RTX 4090 |      NF4     |     BF16     |  13B |       4.00x       |
> | RTX 4090 |      NF4     |     BF16     |  30B |       3.65x       |
> | RTX 4090 |      NF4     |     BF16     |  65B |       3.81x       |
> | A40      |      NF4     |     BF16     |  7B  |       2.94x       |
> | A40      |      NF4     |     BF16     |  13B |       2.91x       |
> | A40      |      NF4     |     BF16     |  30B |       3.02x       |
> | A40      |      NF4     |     BF16     |  65B |       3.25x       |
>
> **Q2: The authors could estimate the potential savings in energy consumption and carbon emissions that could be achieved by using their method to serve LLMs in 4-bit rather than 8-bit or 16 bit.**
>
> **A2**: Please see A1. The estimation of environmental and cost reduction if 50% of deployments are personal deployments is about 72% using our method.
>
> **Q3: They could also discuss the potential impact of their work on the economics of LLMs. For example, if the serving cost is significantly reduced, it could make LLMs more affordable for businesses and organizations.**
>
> **A3**: A main pain point for businesses is (a) efficient deployment with large batch sizes, (b) deploy many different models. Since with QLoRA we can deploy many different finetuning models with the same quantized base model we increase the cost efficiency of (b) very significantly. The cost reductions in (b) is proportional to the number of different adapters deployed (100x reductions are possible with the right implementations). Currently, we do not offer cost reductions for (a), but this can be achieved with the right CUDA optimizations. See A1 for further discussion.
>
> **Q4: Finally, the authors could discuss the challenges that still need to be addressed in order to make their method practical for large-scale deployment.**
>
> **A4**: The main challenge is to make 4-bit matrix multiplication with NF4 viable for large batch sizes. This is possible, but unlike Int4 matrix multiplication, the main bottleneck is instruction throughput on the GPU. We are currently implementing CUDA kernels for large batch sizes (=16) that overlap memory transfers and computation to alleviate instruction throughput bottlenecks. First results yield speedups of ~1.7 - 2.3x for batch size 16, but further improvements are possible. As such, we are confident that we can achieve efficient large-scale deployments of QLoRA models with the right implementations.
>
> **Q5: Can you please replace figure-3 with the table of numbers for Winogrande, HellaSwag, PiQA, Arc-Easy, and Arc-C?**
>
> **A5**: Thank you for this suggestion. We are currently short on space and will include a table that breaks down performance in the appendix.
>
> **Q6: Can you please include the ablation studies on the learning-rate / schedule /hyperparameters that you have done in the appendix? Do all the finetuning experiments shared the same recipt?**
>
> **A6**: We are happy to include all our ablation studies. To ensure we compare against a very strong baseline for full finetuning we first did an extensive hyperparameter search to find the strongest baseline. We found that these baseline parameters are optimal for most models and only learning rate and batch size need to be changed with scale. We also find 16-bit LoRA baselines have the same “best” hyperparameters as 4-bit and 8-bit QLoRA.

---

> > ### Comment · Reviewer_hHMr · 2023-08-17
> > **Thank you**
> >
> > Beautiful responses. Increase my score to 9

---

### Author Rebuttal · Authors · 2023-08-08

We thank all reviewers for their time and contributions. Instead of uploading a document, we inserted relevant tables in markdown in each rebuttal for better readability.

The response has been overwhelmingly positive. We thank the reviewers for their encouragement. We, too, believe that our work will have a groundbreaking effect in making large language model fine-tuning vastly more accessible without compromising performance.

Please let the AC know if you think our work deserves to be highlighted at the conference. We believe many researchers can benefit from learning and using our work, as it enables research even with few GPU resources, which would otherwise be impossible in academic settings. Our work enables research that has not been possible before, particularly for academic researchers with the least resources.

---

### Decision · Program_Chairs · 2023-09-21

**Decision:**

Accept (oral)

**Comment:**

Meta Review for QLoRA: Efficient Finetuning of Quantized LLMs

As diligently summarized by reviewer hHMr, this paper proposes a novel way to finetune 4-bit LLMs for efficient finetuning and serving. QLORA, the proposed method, uses a number of innovations to save memory without sacrificing performance. The authors show that QLORA can be used to finetune a 65B parameter model on a single 48GB GPU, and that it performs well in academic and instruction finetune datasets.

Reviewers, like pPZQ, agree that the main strength lies in the system this paper provides. Quantization and adapter fine-tuning are both standard techniques when it comes to efficiency. However, this paper opens up the possibility of fine tuning a large language model under reasonable cost. Such a possibility is important for the community to study large language models and pursue new research ideas.
Most reviewers, including myself, believe QLORA to be a significant contribution to the field of LLM finetuning and quantization, an area that is right at the spotlight of generative AI, especially in the open-source community where such innovation will be highly impactful as we get our models smaller, more efficient, and to have more capabilities with less resources.

This is a significant work, which deserves to be nominated for an Oral Presentation at NeurIPS.